# Effects of the Feeding Solution Composition on a Reductive/Oxidative Sequential Bioelectrochemical Process for Perchloroethylene Removal

Edoardo Dell'Armi [1], Marco Zeppilli [1,*], Bruna Matturro [2], Simona Rossetti [2], Marco Petrangeli Papini [1] and Mauro Majone [1]

1 Department of Chemistry, Sapienza University of Rome, P.le Aldo Moro 5, 00185 Rome, Italy; edoardo.dellarmi@uniroma1.it (E.D.); marco.petrangelipapini@uniroma1.it (M.P.P.); mauro.majone@uniroma1.it (M.M.)

2 Water Research Institute (IRSA-CNR), Via Salaria km 29.300, 00015 Monterotondo, Italy; matturro@irsa.cnr.it (B.M.); rossetti@irsa.cnr.it (S.R.)

* Correspondence: marco.zeppilli@uniroma1.it; Tel.: +39-0649913716; Fax: +39-06490631

**Abstract:** Chlorinated aliphatic hydrocarbons (CAHs) are common groundwater contaminants due to their improper use in several industrial activities. Specialized microorganisms are able to perform the reductive dechlorination (RD) of high-chlorinated CAHs such as perchloroethylene (PCE), while the low-chlorinated ethenes such as vinyl chloride (VC) are more susceptible to oxidative mechanisms performed by aerobic dechlorinating microorganisms. Bioelectrochemical systems can be used as an effective strategy for the stimulation of both anaerobic and aerobic microbial dechlorination, i.e., a biocathode can be used as an electron donor to perform the RD, while a bioanode can provide the oxygen necessary for the aerobic dechlorination reaction. In this study, a sequential bioelectrochemical process constituted by two membrane-less microbial electrolysis cells connected in series has been, for the first time, operated with synthetic groundwater, also containing sulphate and nitrate, to simulate more realistic process conditions due to the possible establishment of competitive processes for the reducing power, with respect to previous research made with a PCE-contaminated mineral medium (with neither sulphate nor nitrate). The shift from mineral medium to synthetic groundwater showed the establishment of sulphate and nitrate reduction and caused the temporary decrease of the PCE removal efficiency from 100% to 85%. The analysis of the RD biomarkers (i.e., *Dehalococcoides mccartyi* 16S rRNA and tceA, bvcA, vcrA genes) confirmed the decrement of reductive dechlorination performances after the introduction of the synthetic groundwater, also characterized by a lower ionic strength and nutrients content. On the other hand, the system self-adapted the flowing current to the increased demand for the sulphate and nitrate reduction, so that reducing power was not in defect for the RD, although RD coulombic efficiency was less.

**Keywords:** reductive dechlorination; oxidative dechlorination; bioelectrochemical systems; bioremediation

## 1. Introduction

Chlorinated aliphatic hydrocarbons (CAHs), such as perchloroethylene (PCE) and trichloroethylene (TCE), are organic molecules in which chlorine atoms are directly linked to the carbon skeleton. Over the past years, due to the peculiar chemical and physical proprieties, the CAHs were widely utilized in many industrial activities (e.g., dry cleaning devices, degreasing in the heavy industries) [1]. Due to their large use coupled with past uncontrolled handling and disposal, CAHs became ubiquitous contaminants in groundwater, soils, and sediments [2]. Contaminated groundwaters were first treated with chemical–physical treatments; however, in situ biological approaches emerged as an effective strategy, especially for the treatment of the residual contamination. CAHs bioremediation can be performed through the anaerobic reductive pathway, in which hydrogen provides the

reducing power for the CAHs reduction, or through the aerobic oxidative pathway, in which oxygen is used as electron acceptor for the CAHs oxidation. The reductive dechlorination (RD) reaction consists of a microbial reaction in which the ethene skeleton loses a chlorine atom by sequential steps. Each RD step consumes two electrons and provokes the substitution of a chlorine with a hydrogen to the ethene skeleton, i.e., four reductive steps (eight electrons) are required to convert PCE to ethylene (Eth). Several microorganisms are capable of partially degrading PCE and TCE to 1,2-cis-Dichloroethylene (cisDCE) [3,4], while only the *Dehalococcoides mccartyi* (*D. mccartyi*) can perform the total conversion of PCE to ethylene (Eth), the final nontoxic RD byproduct. *D. mccartyi* activity is driven by strain-specific enzymes coded by reductive dehalogenase genes, including *tceA, bvcA,* and *vcrA* [5,6].

Indeed, *Dehalococcoides mccartyi* is the most relevant organohalide respiring microorganism capable of complete RD all the way to ethene through the enzymatic activity of reductive dehalogenase genes [7]. The monitoring of RD biomarkers, often assessed via quantitative PCR (qPCR) including *D. mccartyi* 16S rRNA and reductive dehalogenase genes, is an important tool for the evaluation of the ongoing bioremediation processes, both at laboratory and field scale [8,9].

Being the last RD step from VC to Eth is often cometabolic, CAHs bioremediation can lead to transient accumulation of VC, the more toxic RD byproduct [10], increasing the toxicity of the contamination. In this context, the aerobic degradation of low-chlorinated ethenes can overcome the VC accumulation by its degradation under aerobic conditions [11–13]. The oxidative dechlorination of low-chlorinated cisDCE and VC proceeds through the oxygenase enzyme, which produces chlorinated epoxides by the carbon double bond that easily degrade into $CO_2$ due to their instability [14].

Hence, an effective bioremediation approach is offered by the combination of the reductive and oxidative dechlorination [15], i.e., funnel and gate coupled with oxygen biosparging [16] or nutrient injection with the biosparging [17]. An interesting approach preventing the insertion of chemicals directly in the aquifer is offered by bioelectrochemical systems which involve the interaction between microorganism and a polarized electron [18–21]. In this context, a bioelectrochemical reductive/oxidative sequential process was developed by using a continuous-flow reactor where cathodic and anodic compartments were electrically separated by an ion exchange membrane while sequentially fed with a TCE-contaminated medium (i.e., the cathode was fed first and the effluent from the cathode was fed to the anode) [22]. As a further advancement, a new configuration was recently proposed where the system was split into two membrane-less independent reactors, respectively, for reductive and oxidative reaction, each one having its own counter electrode. This approach makes it possible to independently tune electrical conditions for each reaction, and the absence of separation membrane makes reactor realization simpler and cheaper, especially in a scaling-up perspective [23,24].

In this experimental study, the PCE-contaminated feeding solution used for the investigation of the latter process has been shifted from the previously investigated [23,25] mineral medium to a synthetic groundwater. The main novelty here is that the synthetic groundwater was designed based on the anion content of a real contaminated groundwater, i.e., containing sulphate and nitrate species that can introduce competitive processes for the reducing power [25]. Also, the synthetic groundwater had lower ion strength and less nutrients, being factors of possible impact on the process efficiency. Here we evaluate the performances of the sequential bioelectrochemical process and the RD biomarkers under the different feeding conditions tested.

## 2. Materials and Methods

### 2.1. Sequential Bioelectrochemical Process Operation

The sequential bioelectrochemical process consisted of two tubular microbial electrolysis cells named reductive and oxidative reactor, respectively. As reported in Figure 1, both reductive and oxidative reactor adopted a membrane-less configuration with an internal

graphite counter-electrode. More in detail, the internal graphite counterelectrode consisted of graphite granules enveloped in a plastic mesh which avoided the electric shortcut while allowing the electrolyte migration. The working electrode of the reductive reactor was also constituted by graphite granules, while in the oxidative reactor, the working electrode was a commercial mixed metal oxides (MMO) electrode (Magneto Special Anodes, Schiedam, The Netherland).

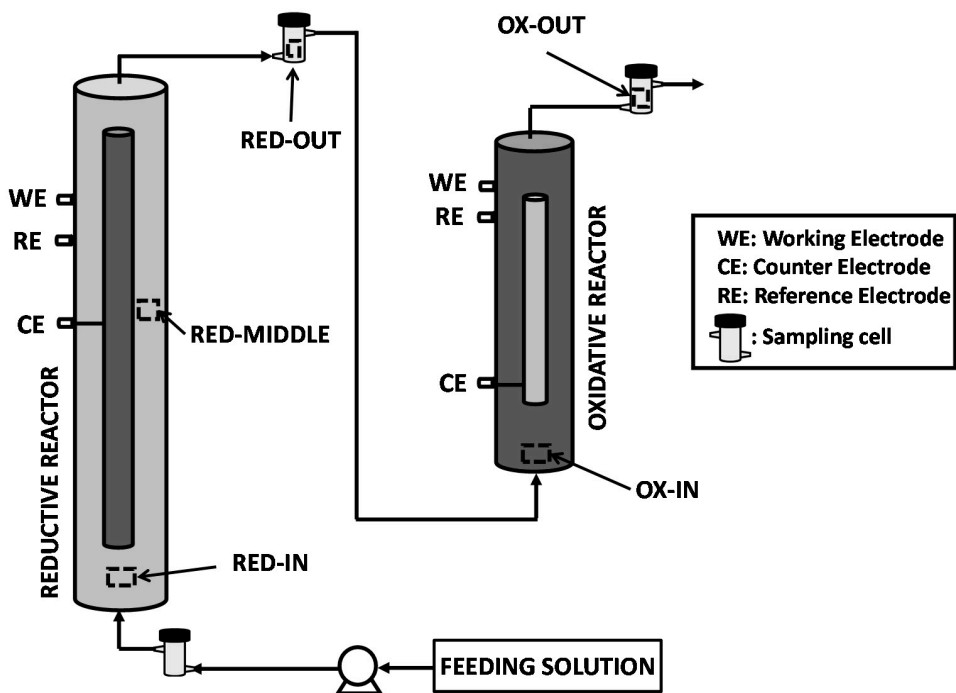

**Figure 1.** Sequential reductive/oxidative bioelectrochemical process scheme; reprinted with the permission of Elsevier.

The mineral medium solution utilized as feeding solution in the first operational period was the same as previously reported [23]. The synthetic groundwater feeding solution consisted of a tap water solution whose composition is reported in Table 1.

**Table 1.** Synthetic groundwater solution composition.

| Compound | Concentration (mg/L) |
|---|---|
| Tap water | - |
| $Na_2SO_4$ | 680 |
| $NaNO_3$ | 20 |

Both feeding solutions were contaminated with a theoretical perchloroethylene (PCE) concentration of 100 μM and fed to the sequential process by a peristaltic pump. The reductive reactor was polarized with the cathode (i.e., the working electrode) at −450 mV vs. standard hydrogen electrode (SHE), while the oxidative reactor anode was polarized at +1.4 V vs. SHE.

### 2.2. Analytical Methods

The CAHs, ethylene, ethane, and methane were detected as already described, by sampling cell headspace injection in a DANI-MASTER gas chromatograph (DANI Instruments, Contone, Switzerland). CAHs, ethylene, and ethane were detected using a flame ionization detector (FID), while $H_2$ and $CH_4$ were analyzed by a thermal conductivity detector (TCD). The aqueous-phase concentrations were calculated, in liquid–gas equilibrium condition

assumptions, by converting the headspace gaseous concentrations using tabulated Henry's law constants [26]. The sulphate and nitrate were analyzed by injecting filtered samples in a Dionex ICS-1000 (Dionex, Sunnyvale, CA, USA) ion chromatograph equipped with a conductivity detector as described in [25].

### 2.3. D. mccartyi Quantification

Real-time quantification (qPCR) was conducted on DNA extracted from 5 mL of liquid effluent taken from the outlet of the reductive and oxidative reactors. Samples were collected at the end of the operating periods conducted with mineral medium and with the synthetic groundwater. Liquid samples were filtered on polycarbonate filters (pore size 0.22 μm, 47 mm diameter, Nuclepore) to harvest the biomass. DNA was extracted directly from the filters by PowerSoil DNA extraction kit (Qiagen, Milan, Italy) following the manufacturer's instructions. Purified DNA from each sample was eluted in 100 μL sterile Milli-Q for qPCR absolute quantification of *D. mccartyi* 16S rRNA gene and reductive dehalogenase genes tceA, bvcA, and vcrA. Reactions were conducted in 20 μL total volume including 3 μL of DNA as template, 300 nM of each primer, and TaqMan® probe and SsoAd-vancedTM Universal Probes Supermix (Bio-Rad, Segrate, Italy) as previously reported [24]. Quantitative data were reported as gene copy numbers L-1 of liquid medium.

### 2.4. Calculations

The main parameters already described in [23,24] are summarized in Table 2.

**Table 2.** Parameters for the description of reductive and oxidative processes.

| | |
|---|---|
| **PCE removal** <br> **(μmol/Ld)** | $PCE\,removal\,(\mu mol/Ld) = [PCE]_{in} - [PCE]_{out} \times Q_{liquid}/V_{reductive}$ |
| **Reductive dechlorination rate** <br> **(RD, μeq/Ld)** | $RD\,(\mu eq/Ld) = Q_{liquid}/V_{reductive} \times [TCE] \times 2 + [cisDCE] \times 4 + [VC] \times 6 + [Eth] \times 8 + [Eta] \times 10$ <br> $RD\,(mA) = RD\,(\mu eq/Ld) \times V_{reductive} \times F/86,400/1000$ |
| **Methane** <br> **production rate** <br> **(rCH$_{4(eq)}$)** | $RCH_4\,(\mu eq/Ld) = Q_{liquid}/V_{reductive} \times [CH_4] \times 8$ <br> $RCH_4\,(mA) = RCH_4\,(\mu eq/Ld) \times V_{reductive} \times F/86,400/1000$ |
| **Sulphate (RD) and Nitrate (RN) removal rate** <br> **(RS, RN$_{(μeq/Ld)}$)** | $RS\,(\mu eq/Ld) = Q_{liquid}/V_{reductive} \times [SO_4^{-2}]_{removed} \times 8$ <br> $RS\,(mA) = RS\,(\mu eq/Ld) \times V_{reductive} \times F/86,400$ <br> $RN\,(\mu eq/Ld) = Q_{liquid}/V_{reductive} \times [NO_3^-]_{removed} \times 5$ <br> $RN\,(mA) = RN\,(\mu eq/Ld) \times V_{reductive} \times F/86,400$ |
| **Oxidative dechlorination rate** <br> **(OD, μmol/Ld)** | $OD(CAHs)\,(\mu mol/Ld) = Q_{liquid}/V_{reductive} \times [CAHs]_{in} - [CAHs]_{out}$ <br> $OD(CAHs)\,(mmolO_2/d) = OD(CAHs)\,(\mu mol/Ld) \times V_{oxidative} \times \alpha/1000$ <br> $\alpha_{VC} = 2.5\ \alpha_{Eth} = 3\ \alpha_{Eta} = 3.5\ \alpha_{CH4} = 2$ |
| **RD Coulombic Efficiency** <br> **(CE$_{RD}$, %)** | $CE_{RD} = RD\,(mA)/I_{redcutive}\,(mA) \times 100$ |
| **RS Coulombic Efficiency** <br> **(CE$_{RS}$, %)** | $CE_{RS} = RS\,(mA)/I_{redcutive}\,(mA) \times 100$ |
| **RN Coulombic Efficiency** <br> **(CE$_{RN}$, %)** | $CE_{RN} = RN\,(mA)/I_{redcutive}\,(mA) \times 100$ |
| **RCH4 Coulombic Efficiency** <br> **(CE$_{CH4}$, %)** | $CE_{CH4} = RCH_4\,(mA)/I_{redcutive}\,(mA) \times 100$ |
| **Oxidative coulombic efficiency** <br> **(CE$_{OD}$, %)** | $I_{oxidative}\,(mmolO_2/d) = I_{oxidative}\,(mA)/4/F \times 86,400$ <br> $CE_{OD} = OD(CAHs)\,(mmolO_2/d)/I_{oxidative}\,(mmolO_2/d) \times 100$ |
| | F = 96,485 C/mol e$^-$　86,400 = s/d　Q$_{liquid}$ = liquid flow rate　Q$_{gas}$ = gaseous flow rate |

## 3. Results

*3.1. Continuous Flow Operation of the Sequential Bioelectrochemical Process with the Mineral Medium*

After previous experiments devoted to investigating the reductive potential effect [24], here, the reductive reactor was polarized with a potential of –450 mV vs. SHE during the entire experimental period. The sequential reductive/oxidative process was operated first with the mineral medium to achieve a stable operation, by using an average flow rate of $2.2 \pm 0.7$ L/d, which resulted in an average hydraulic retention time (HRT) of 3.7 days for the reductive reactor. As shown in Figure 2a, PCE was completely removed with an average rate of $26 \pm 9$ μmol/Ld, corresponding to a removal efficiency of $99 \pm 1\%$. The PCE dechlorination products detected in reductive reactor effluent were predominantly vinyl chloride (VC) ethylene (Eth) and ethane (Eta), i.e., no high-chlorinated byproducts were detected in the reductive reactor effluent (Figure 2b). The average concentration of VC, Eth, and Eta resulted in $55 \pm 5$, $20 \pm 3$, and $12 \pm 1$ μmol/L, respectively (Figure 2c). The average current recorded in the reductive reactor resulted in $-19.3 \pm 0.3$ mA, while the applied cell voltage and the counter electrode potential resulted in $-1.4 \pm 0.1$ V and $+0.8 \pm 0.1$ V vs. SHE, respectively. The RD reaction reached an average rate of $177 \pm 8$ μeq/Ld, corresponding to an average coulombic efficiency of $8 \pm 1\%$, a similar value obtained under the same operating conditions [23].

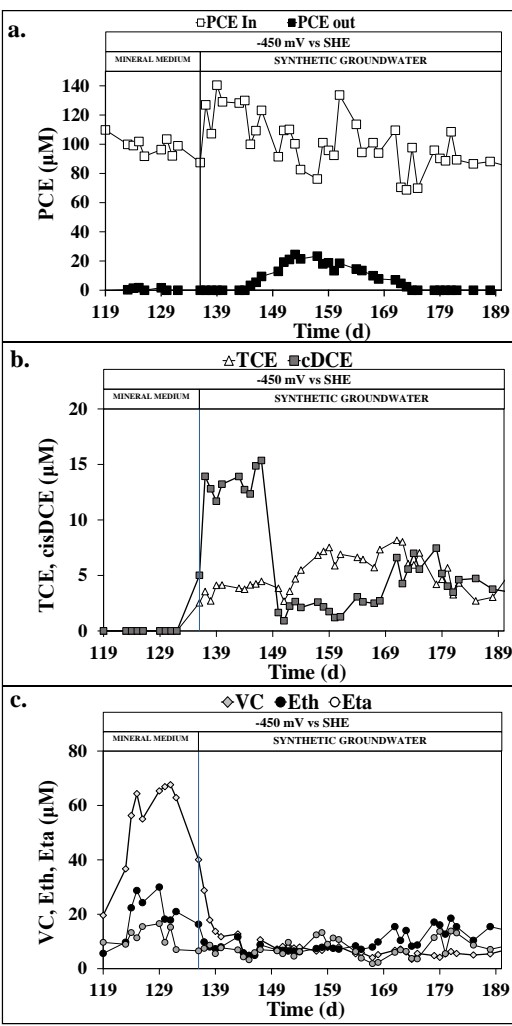

**Figure 2.** Perchloroethylene (PCE) concentration in the influent and effluent of the reductive reactor (**a**), high-chlorinated (**b**) and low-chlorinated (**c**) reductive dechlorination byproducts in the effluent of the reductive reactor during the continuous flow runs with mineral medium and synthetic groundwater.

Methane production (Figure 3) through bioelectromethanogenesis reaction [27] accounted for the consumption of the $2 \pm 1\%$ of the reactor flowing current (methane coulombic efficiency), corresponding to an average methane production rate of $5.5 \pm 0.7$ µmol/Ld. The overall current recovery reached only $10 \pm 2\%$, i.e., a considerable amount of electric current resulted in unrecovered by using the internal counter electrode and a membrane-less configuration. Therefore, as already reported by other authors [28], the presence of electrons loops of species such as hydrogen (which can be reduced and oxidized in the different reactors part) usually promotes a scavenging effect of the available equivalents with a consequent low coulombic efficiency.

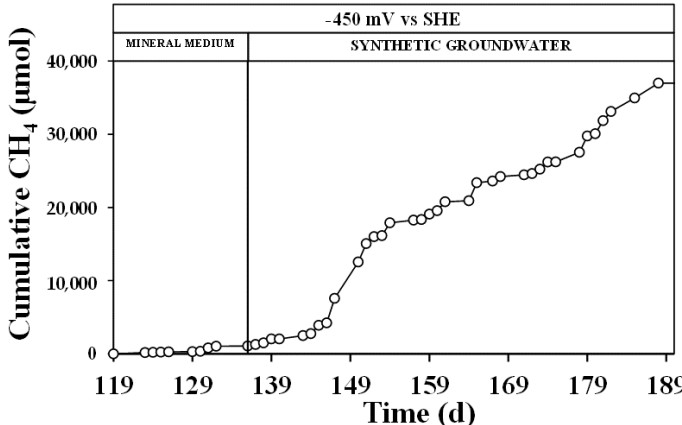

**Figure 3.** Cumulative methane in the reductive reactor for the continuous flows march at –450 mV vs. standard hydrogen electrode (SHE) with mineral medium and synthetic groundwater, respectively.

The reductive reactor outlet constituted the inlet of the oxidative reactor which was operated with the working electrode (i.e., the anode) polarized at +1.4 V vs. SHE to ensure $O_2$ evolution necessary for the oxidative dechlorination pathways [29]. By operating with an average HRT of 1.6 d, the oxidative reactor performed the almost complete removal of VC (Figure 4a), Eth (Figure 4b), and Eta (Figure 4c) with average removal efficiencies of $94 \pm 2\%$, $98 \pm 5\%$, and $100 \pm 2\%$, respectively.

The VC, Eth, and Eta removal confirmed the results obtained under the same operating condition [23]. Moreover, with an average oxidative current of $8.0 \pm 0.3$ mA, the oxidative coulombic efficiency for the VC, Eth, and Eta accounted for $16 \pm 3$, $7 \pm 1$, and $4 \pm 1\%$, respectively. The methane produced in the reductive rector was oxidized with an average removal efficiency of $65 \pm 8\%$ in the oxidative reactor, with a corresponding oxidative coulombic efficiency of $3 \pm 2\%$.

### 3.2. Continuous Flow Operation of the Sequential Bioelectrochemical Process with the Synthetic Groundwater

After the shift to the synthetic groundwater feeding, a partial loss of the PCE removal was observed after one reductive reactor HRT, i.e., after 4 days, a PCE concentration up to $20 \pm 5$ µmol/L was detected in the reductive effluent (Figure 2a) which clearly indicates the decrease of activity of the dechlorinating biomass. The PCE removal rate dropped down to $21 \pm 2$ µmol/Ld during the shock period, corresponding to a removal efficiency of $85 \pm 2\%$. During the transient period, the presence of high-chlorinated RD products was highlighted in the reductive reactor effluent, confirming the dechlorinating biomass shock due to the feeding solution change (Figure 2b). After almost two additional HRT, the PCE removal capacity of the reductive reactor was completely restored with a complete PCE removal. Even if the PCE removal was restored, the RD products concentration detected in the PCE byproducts remained similar to the shock period of the dechlorinating biomass with average concentrations of $4 \pm 1$, $4 \pm 1$, $6 \pm 1$, $8 \pm 1$, and $13 \pm 2$ µmol/L, for TCE, cisDCE, VC, Eth, and Eta, respectively. The global RD rate reached a value of $68 \pm 9$ µeq/Ld, which

corresponded to an average coulombic efficiency of $1 \pm 1\%$. Table 3 reports the main parameters regarding the RD reaction in the reductive reactor.

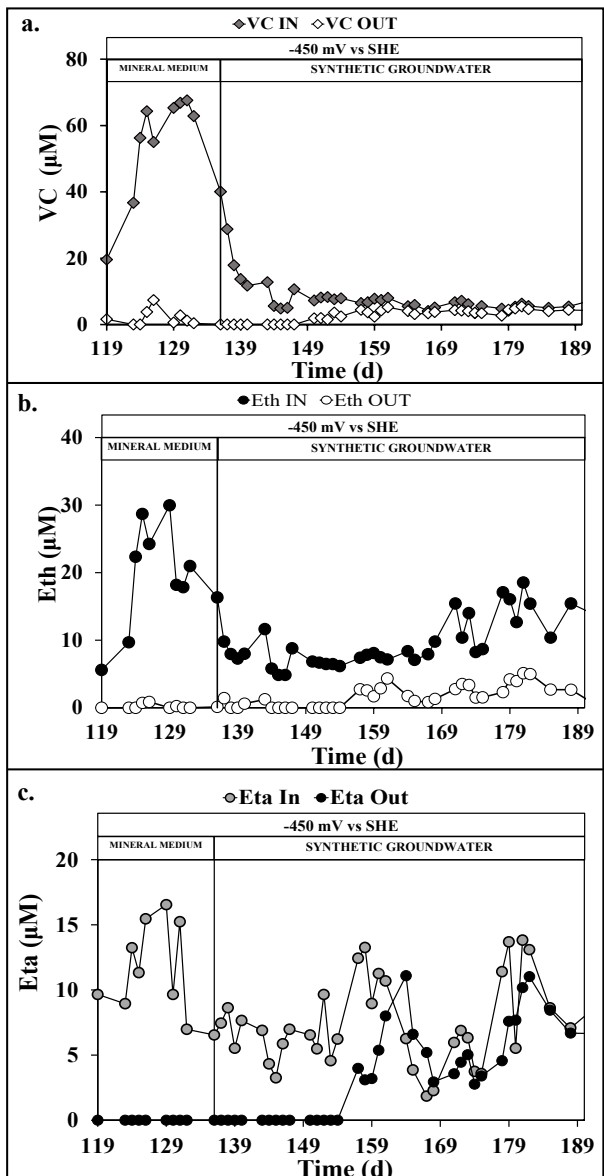

**Figure 4.** Vinyl chloride (VC) (**a**), Ethylene (**b**), and Ethane (**c**) in the oxidative reactor during the continuous flow runs with mineral medium and synthetic groundwater.

**Table 3.** Reductive reactor performances with the two feeding solutions.

| Reductive Reactor HRT 3.9 d @ −450 mV vs. SHE | | |
|---|---|---|
| **Feeding Solution** | **Mineral Medium** | **Synthetic Groundwater** |
| PCE removal rate (μmol/Ld) | $26 \pm 9$ | $22 \pm 2$ |
| PCE removal efficiency (%) | $99 \pm 4$ | $95 \pm 6$ |
| RD Rate (μeq/Ld) | $177 \pm$ | $68 \pm 1$ |
| Current (mA) | $-19 \pm 1$ | $-65 \pm 3$ |
| RD Coulombic Efficiency (%) | $8 \pm 2$ | $1 \pm 1$ |

As reported in Figure 5, the synthetic groundwater introduction in the sequential bioelectrochemical process resulted in a sharp current increase after one HRT, i.e., the average current value increased from $-17.8 \pm 0.6$ mA to $64.6 \pm 2.9$ mA. According to the

current increase, the cell voltage applied between the cathode and the anode increased from $-1.71 \pm 0.02$ to $-4.11 \pm 0.05$ V. The cell voltage increase was related both to the current increase and by the lower synthetic groundwater conductivity ($0.83 \pm 0.02$ vs. $2.68 \pm 0.11$ mS/cm), which increased the ohmic resistance of the cell. The current increase was promoted by the presence of other reduction reactions, which caused the increase in reducing power consumption at the potential of $-450$ mV vs. SHE.

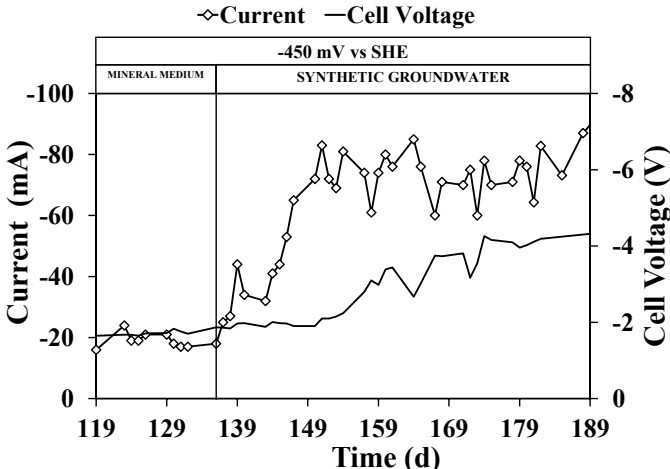

**Figure 5.** Current and cell voltage time course in the reductive reactor during the continuous flow runs with mineral medium and synthetic groundwater.

Indeed, the presence of sulphate and nitrate in the influent synthetic groundwater promoted the establishment of the sulphate reduction and nitrate reduction reactions in the reductive reactor. As shown in Figure 6a, the average influent and effluent concentration of sulphate was $434 \pm 17$ and $279 \pm 15$ mgSO$_4{}^{-2}$/L, which corresponds to a daily removal of $0.39 \pm 0.08$ mmol/Ld of sulphate. By the hypothesis of the complete reduction of sulphate to sulphide, frequently reported in bioelectrochemical systems [30,31], a daily reducing power consumption of $3.2 \pm 0.5$ meq/Ld for the sulphate reduction reaction was observed, which accounted for $45 \pm 9\%$ of the current flowing in the reductive reactor, i.e., the sulphate reduction coulombic efficiency. In a similar way, as reported in Figure 6b, the complete nitrate reduction [32,33] promoted the consumption of $0.3 \pm 0.1$ meq/Ld of nitrate, which accounted for $4 \pm 1\%$ of the average current flowing in the reductive reactor.

The higher average flowing current also promoted the increase of methane production through the bioelectromethanogenesis reaction, as reported in the literature [34]. Indeed, the methane production during the synthetic groundwater operation accounted for an average methane production of $94 \pm 24$ μmol/Ld, which in turn contributed for the $11 \pm 5\%$ of the global flowing current in the reductive reactor. Considering the synthetic groundwater period, a global current recovery of the $61 \pm 16\%$ was obtained by the different bioelectrochemical reduction reaction, summarized in Table 4.

**Table 4.** Coulombic efficiencies for the different reduction processes detected in the reductive reactor during the operation with the synthetic groundwater.

| Reductive Reactor HRT 3.9 d $-450$ mV vs. SHE | |
| --- | --- |
| Coulombic Efficiency RD (%) | $1 \pm 1$ |
| Coulombic Efficiency Sulphate Reduction (%) | $45 \pm 9$ |
| Coulombic Efficiency Nitrate Reduction (%) | $4 \pm 1$ |
| Coulombic Efficiency CH$_4$ (%) | $11 \pm 5$ |
| Global Coulombic Efficiency (%) | $61 \pm 16$ |

Although the different global coulombic efficiencies were reached by the reductive reactor during the two operating periods, with average values of $10 \pm 2$ and $61 \pm 16\%$, the unrecovered current, i.e., the current that was not justified by the identified reduction reactions, reached a similar value. Indeed, while 17 mA (i.e., 90% of 19 mA) resulted as unrecovered during the first mineral medium period, during the synthetic groundwater period, the unrecovered current reached 25 mA (39% of 65 mA). Based on this electron balance consideration, it is possible to underline that the dechlorinating biomass shock was not related to the nitrate and sulphate reduction reaction's competition for the reducing power. The decrease of the RD rate was probably related to the different composition of the feeding solution, both in terms of nutrients and ionic strength, i.e., while the mineral medium conductivity resulted in, on average, $2.68 \pm 0.11$ mS/cm, the synthetic groundwater conductivity was $0.83 \pm 0.02$ mS/cm, according to this consideration. In addition, a previous experiment is reported in the literature [25] in which the bioelectrochemical RD stimulation on cisDCE in presence of sulphate and nitrate showed that most of the current was consumed for anions reduction and the methane production. Moreover, the presence of nitrate and sulphate reduction reactions did not result in a competition for the reducing power with the RD reaction but in an increase of the reducing power availability in the circuit, i.e., the current increase.

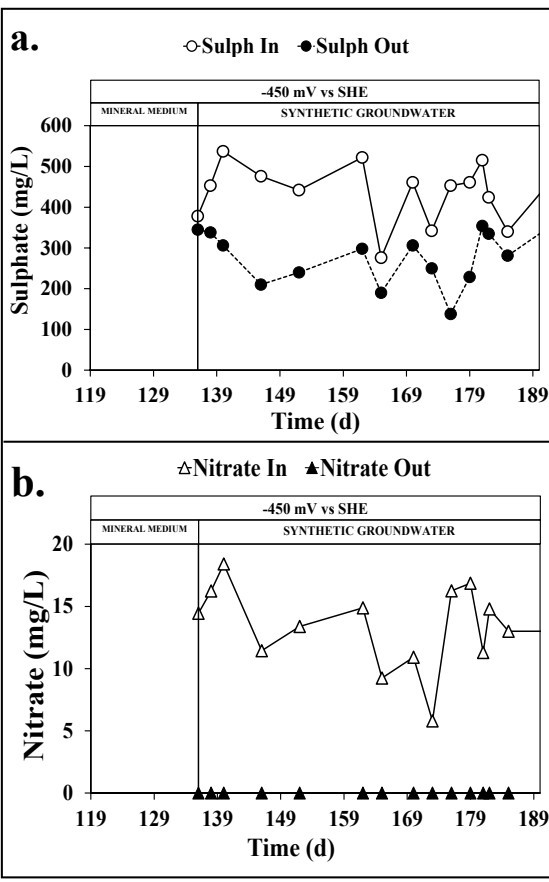

**Figure 6.** Sulphate removal (**a**) and nitrate removal (**b**) in the reductive reactor during the continuous flow runs with mineral medium and synthetic groundwater.

As reported in Figure 4B, the oxidative reactor performed an almost complete Eth removal, with an average removal efficiency of $81 \pm 11$. On the contrary, after the introduction of the synthetic groundwater feeding solution, the VC and Eta removal efficiency decreased. More in detail, as shown in Figure 4a,c, after 15 days from the introduction of the synthetic groundwater feeding in the oxidative reactor (almost 10 HRT), the progressive decrease of the VC and Eta removal efficiency from $96 \pm 2\%$ to $36 \pm 5\%$ and from

100 ± 2 to 31 ± 8% was observed, respectively. The loss of VC and Eta removal efficiency corresponded to the introduction of highly chlorinated compounds, such as PCE, TCE, and cisDCE coming from the incomplete RD in the reductive reactor (Figure 7). This likely resulted in the progressive acclimation of RD pathways also in the oxidative reactor. As reported in Figure 7a, the influent PCE resulted slightly removed in the oxidative reactor, while in Figure 7b, the 86 ± 3% of the influent TCE was removed. Even if TCE removal can also proceed by a cometabolic aerobic mechanism in presence of methane [35], the considerable increase of cisDCE concentration reported in Figure 7c confirmed the presence of the RD in the oxidative reactor. The presence of the RD was easily explained by the continuous inoculation of dechlorinating anaerobic bacteria coming from the reductive reactor that get acclimated in the internal counter electrode of the oxidative reactor, which acted as electron donor thanks to its average potential of −0.7 ± 0.1 V vs. SHE. Finally, considering the RD of PCE and TCE to cisDCE, an average RD of 116 ± 32 µeq/Ld was evaluated, which corresponded to an RD coulombic efficiency of 2 ± 1% for the oxidative reactor. On the contrary, considering the aerobic dechlorination pathway on VC, Eth, and Eta, the oxidative coulombic efficiencies resulted in 1 ± 1, 5 ± 2, and 1 ± 1%, respectively.

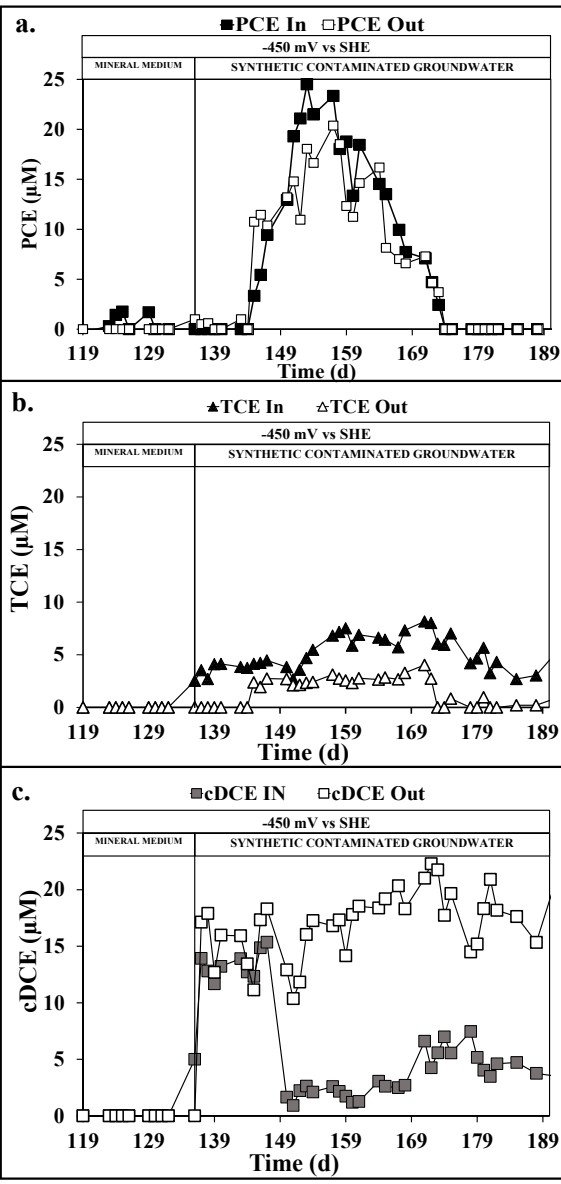

**Figure 7.** PCE (**a**), trichloroethylene (TCE) (**b**), and 1,2-cis-Dichloroethylene (cisDCE) (**c**) in the oxidative reactor during the continuous flow runs with mineral medium and synthetic groundwater.

### 3.3. Biomarkers Investigation under the Two Operating Periods

The main biomarkers determined at the end of the reductive and oxidative reactors operation included *D. mccartyi* 16S rRNA and *tceA*, *bvcA*, *vcrA* reductive dehalogenase genes. *D. mccartyi* was found both in the reductive reactor fed with the mineral medium ($1.30 \times 10^8$ 16S rRNA gene copies/L) and with the synthetic groundwater ($2.92 \times 10^6$ 16S rRNA gene copies/L) (Figure 8). *tceA*-carrying strains represented the majority of *D. mccartyi* detected in all samples analyzed (Table 5). At a lower extent, the occurrence of *vcrA* gene ($\leq 2.89 \times 10^3$ gene copies/L) was also found at the outlet of the reductive reactors.

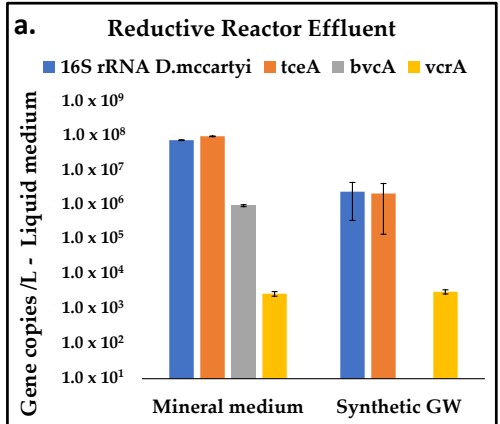 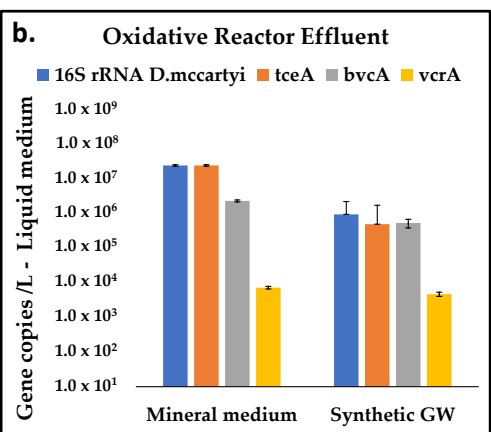

**Figure 8.** Gene copies concentration with the two feeding solutions, in the reductive reactor effluent (**a**), and in the oxidative reactor effluent (**b**).

**Table 5.** Biomarkers gene copy numbers per L of mineral medium or synthetic groundwater.

|  |  | 16S rRNA *D. mccartyi* | *tceA* | *bvcA* | *vcrA* |
|---|---|---|---|---|---|
| **Mineral Medium** | **RR * Effluent** | $1.30 \times 10^8$ | $1.73 \times 10^8$ | $1.91 \times 10^6$ | $2.01 \times 10^3$ |
|  | **OR * Effluent** | $2.34 \times 10^7$ | $2.33 \times 10^7$ | $2.23 \times 10^6$ | $7.12 \times 10^3$ |
| **Synthetic Groundwater** | **RR * Effluent** | $2.92 \times 10^6$ | $2.89 \times 10^6$ | $0.00 \times 10^1$ | $2.89 \times 10^3$ |
|  | **OR * Effluent** | $9.10 \times 10^5$ | $4.76 \times 10^5$ | $5.13 \times 10^5$ | $4.60 \times 10^3$ |

\* Reductive Reactor (RR); Oxidative Reactor (OR).

In line with the biomarkers characterization previously performed during the operation with mineral medium of the reductive reactor at $-550$ mV vs. SHE [24], the *D. mccartyi* 16S rRNA and *tceA*, *bvcA*, *vcrA* reductive dehalogenase genes abundance resulted in almost constant, showing a not significant difference in terms of gene/copies in the liquid medium of the reductive reactor.

Consistent with the different composition of the dechlorination products observed under the two feeding conditions, a marked reduction of *D. mccartyi* carrying the reductive dehalogenase *bvcA* involved in VC reductive dechlorination was found in the system fed with synthetic groundwater (Figure 8a). Moreover, in line with RD rates estimated under the two feeding conditions, *D. mccartyi* and *tceA* abundances decreased by at least two orders of magnitude when the synthetic groundwater was used as feeding solution (Figure 8a). This decrement can be likely attributed to the different ionic strength of the two feeding solutions, along with the absence of trace nutrients in the synthetic groundwater which may affect the dechlorination performances.

*D. mccartyi* and the relative reductive dehalogenase genes were also found in the oxidative reactor (Figure 8b). The presence of *D. mccartyi* in the oxidative reactor was likely due to the presence of the internal graphite counter electrode (i.e., the cathodic chamber of the oxidative reactor) where dechlorinating microorganisms can adopt the graphite granules as an electron donor. This finding supports the reported chemical data which suggested the presence of the RD in the oxidative reactor. Similar to the reductive reactor,

also in the liquid sample collected at the outlet of the oxidative unit, an important decrease of *D. mccartyi* was observed during the operation with both feeding solutions.

## 4. Conclusions

The shift of the feeding solution from a well-balanced mineral medium to a simpler synthetic groundwater caused a temporary inhibition of the PCE removal efficiency, which dropped down from $99 \pm 1$ to $85 \pm 2\%$. In line with less PCE removal, *D. mccartyi* abundances also decreased, probably due to the lower ionic strength and nutrients content of the synthetic groundwater. Even if the PCE complete removal was restored after four reductive reactor HRTs, the introduction of the synthetic groundwater lowered the RD rate from $177 \pm 4$ to $68 \pm 1$ µeq/Ld. The sulphate and nitrate presence in the synthetic groundwater promoted the increase of the average current due to their bioelectrochemical reduction, which accounted for the $45 \pm 12$ and $4 \pm 1\%$ of the current flowing in the reductive reactor, respectively. Moreover, as also confirmed by the analysis of the RD biomarkers in the different reactors of the sequential process, the introduction of high-chlorinated compounds in the oxidative reactor promoted the presence of both oxidative and RD pathways. Even though the system performance decreased when shifting the feeding solution from mineral medium to synthetic groundwater, this decreased performance is, anyway, more representative of true conditions that can be encountered in the field. Besides this, the obtained data showed an interesting perspective for the sequential reductive/oxidative process application for CAHs removal, especially because it was evident that the system self-adapted the flowing current to the increased demand for the sulphate and nitrate reduction, so that reducing power was not in defect for the RD. On the other hand, it was evident that medium composition has a significant effect that requires further investigation. Moreover, a more detailed investigation of the reductive and oxidative reactors operating condition should be addressed for the optimization of applied potential and HRT, thanks to the independent regulation of the electrical and fluidynamic conditions for each reactor. Finally, it is noteworthy that the absence of separation membrane makes reactor realization simpler and cheaper for a scaling-up perspective. The system is a good candidate to address CAH remediation with no need to add any chemical substances to the groundwater.

**Author Contributions:** E.D.—investigation, preparation of the original draft; M.Z.—investigation, preparation of the original draft; B.M.—investigation, preparation of the original draft; M.P.P.—supervision; S.R.—supervision; M.M.—supervision, funding Acquisition. All authors have read and agreed to the published version of the manuscript.

**Funding:** This project has received funding from the European Union's Horizon 2020 research and innovation programme under grant agreement No 826244-ELECTRA.

**Institutional Review Board Statement:** Not applicable.

**Informed Consent Statement:** Not applicable.

**Data Availability Statement:** The data presented in this study are openly available in Zenodo, reference number 10.5281/zenodo.4559008.

**Acknowledgments:** Vladimir Chiacchiarini is acknowledged for his skillful assistance in the experimental activity.

**Conflicts of Interest:** The authors declare no conflict of interest.

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
