# Peer review of "Effects of the Feeding Solution Composition on a Reductive/Oxidative Sequential Bioelectrochemical Process for Perchloroethylene Removal"

_processes, doi:10.3390/pr9030405_

Round 1

Reviewer 1 Report

the manuscript is well written and scientifically sound. I only have minor comments that could further improve the delivery of its contents:

Lines 79-80:" the synthetic groundwater was designed based on the anion content of a real contaminated groundwater," a table with the synthetic GW composition should be included

Lines 88-89: "The sequential bioelectrochemical process consisted in two tubular microbial electrolysis cells (MEC).." authors should include a scheme of the system described..

Line 104: methods description should be reported less synthetically. the idea would be that a reader could understand which methods were used without referring to previously published works. 

Line 122: same comment as before. Without repeating the entire M&M section of the previously published papers, a brief summary could be given here, considering that the manuscript is not overly long. This would make it easier for a reader to get full information without reading a previous paper.

Figure 1.b is too "dense" I suggest splitting the information contained therein in two separate graphs (there are two Y-scales anyway) 1.b and 1.c, which could take the same overall space  of Figure 1 as currently presented.

In Figure 3 the same vertical scale is used in the three graphs. This squeezes the shown values unnecessarily, as they depict different parameters.

Conclusions look more like a results discussion session. Perhaps the section is should be re-labeled as such and a brief paragraph of conclusions added afterwards.

Author Response

Reviewer 1 Comments and Suggestions:

The manuscript is well written and scientifically sound. I only have minor comments that could further improve the delivery of its contents:

Reviewer 1: Lines 79-80:" the synthetic groundwater was designed based on the anion content of a real contaminated groundwater," a table with the synthetic GW composition should be included

Author: According to reviewer 1 suggestion a table with the synthetic groundwater composition has been inserted in section 2.1 of the manuscript.

Reviewer 1: Lines 88-89: "The sequential bioelectrochemical process consisted in two tubular microbial electrolysis cells (MEC).." authors should include a scheme of the system described..

Author: According to reviewer 1 suggestion the scheme of the sequential process has been inserted in section 2.1 of the revised version of the manuscript.

Reviewer 1: Line 104: methods description should be reported less synthetically. the idea would be that a reader could understand which methods were used without referring to previously published works. 

Reviewer 1: Line 122: same comment as before. Without repeating the entire M&M section of the previously published papers, a brief summary could be given here, considering that the manuscript is not overly long. This would make it easier for a reader to get full information without reading a previous paper.

Author: According to reviewer 1 suggestions (Line104 - Line 122) a more detailed description of the analytical methods has been inserted in the revised version of the manuscript.

Reviewer 1: Figure 1.b is too "dense" I suggest splitting the information contained therein in two separate graphs (there are two Y-scales anyway) 1.b and 1.c, which could take the same overall space of Figure 1 as currently presented.

Author: According to reviewer comment, former figure 1-B, has been divided into two figures which report the high-chlorinated (new figure 2-B) and low-chlorinated (new figure 2-C) reductive dechlorination by-products. The figures captions in the text have been accordingly revised.

Reviewer 1: In Figure 3 the same vertical scale is used in the three graphs. This squeezes the shown values unnecessarily, as they depict different parameters.

Author: According to reviewer comment, the vertical scale of the three different graphs of former figure 3 have been optimized for each chlorinated compound.

Reviewer 1: Conclusions look more like a results discussion session. Perhaps the section is should be re-labeled as such and a brief paragraph of conclusions added afterwards.

Author: according to reviewer 1 and reviewer 2 suggestion the conclusion part of the revised version of the manuscript has been improved.

Reviewer 2 Report

See attached.

Author Response

Reviewer 2: The references need to be cleaned up and those that are not referred to in the paper or are barely relevant should be removed. Citing one’s own work, if it is relevant to the paper at hand, is perfectly fine. This is the case with the authors’ reference [23]. However, their work [26] and [27] are referenced only once in passing: Lines 105-106: The CAHs, ethylene, ethane and methane were detected as already described in [25-27]. (emphasis mine) Actually, the authors’ papers [26] and [27] do NOT focus on “CAHs, ethylene, ethane and methane detection.” Similar story with the author’s references [24], [28], [30], and [35]. They should all be removed from the manuscript as they are only marginally, if at all, relevant. You can google “self-citations and ethics” to learn more about the issue.

Author: Reference 26 and 27 give some detailed information about the analytical determination of CH4 and CAHs, however, they are not essential for the paper description and they have been removed from the revised version of the manuscript according to the reviewer suggestion. Regarding the other references cited by the reviewer 2, the explanation of their citation is following reported:

- Reference [24] describes the same experimental set up operated under different operating condition (different cathodic potential with the mineral medium solution) giving information related to the experimental set-up and the experimental results. Moreover, reference [24] reports the microbiological community characterization of the process, which is useful for biomarkers determination reported in the paper.

- Reference [28] and [35] are related to experimental works in which the methane is produced through a biocathode in a similar Microbial Electrolysis cell. More in details reference [35] shows the dependence of methane generation by the average current flowing in the circuit, which is consistent with the experimental results obtained after the introduction of the synthetic groundwater which promoted the current increase.

- Reference [30], finally has been cited because it represents one of the first experimental evidence of the stimulation of an aerobic dechlorinating colture able to oxidize cis DCE by a bioelectrochemical system.

ACRONYMS

Author: All the suggestions related to the ACRONYM use and spelling have been revised according to the reviewer indications:

- minor frequently used acronyms have been eliminated (BES – MEC – NR – SR – Rdase)

- more frequently used acronyms have been checked through the text (HRT – Eth – cisDCE – RD – D.mccarty)

Other comments

Reviewer 2: First, you can do a much better job defining the novel contribution of your paper. And make it clear in the abstract and conclusions.

Author: according to your suggestions the novelty of the paper has been better highlighted in the abstract and conclusion part of the revised version of the manuscript.

Reviewer 2: Second, groundwater is the source of 25-50% of drinking water in USA and close to 100% in some countries. The term “synthetic groundwater” does not imply “synthetic contaminated groundwater,” but it is in this sense that you are using the term. So why not be more precise and replace the term “synthetic groundwater” with “synthetic contaminated groundwater”?

Author: we agree with the reviewer about the importance of underlining the groundwater contamination in the paper, however, the terms utilized for the definition of the feeding solutions, i.e. mineral medium and synthetic groundwater are referred only to the matrixes utilized in the study. Indeed, both feeding solutions were contaminated with the same amount of perchloroethylene as mentioned in line 103-104 of the revised version of the manuscript “both feeding solutions were contaminated with a theoretical perchloroethylene (PCE) concentration of 100 µM”. For this reason, we preferred to maintain the term synthetic groundwater in the revised version of the manuscript.

Reviewer 2: Third, by using the term groundwater and by opening the paper with the sentence “Chlorinated aliphatic hydrocarbons (CAHs) are common groundwater contaminants ...”, you made it relevant to groundwater. What’s missing in the paper is a paragraph or two, or perhaps even a short speculative section, on future research and potential applications of your findings to bioremediation of groundwater contaminated with CAHs.

Author: according to your suggestion some comments related to the future perspective of the process in the frame of groundwater bioremediation have been inserted in the conclusion section of the revised version of the manuscript.
